# Resource Allocation in Spectrum Access System Using Multi-Objective Optimization Methods

**DOI:** 10.3390/s22041318

**Published:** 2022-02-09

**Authors:** Waseem Abbass, Riaz Hussain, Jaroslav Frnda, Nasim Abbas, Muhammad Awais Javed, Shahzad A. Malik

**Affiliations:** 1Department of Electrical and Computer Engineering, COMSATS University, Islamabad 45550, Pakistan; waseem.abbas@comsats.edu.pk (W.A.); rhussain@comsats.edu.pk (R.H.); smalik@comsats.edu.pk (S.A.M.); 2Department of Quantitative Methods and Economic Informatics, Faculty of Operation and Economics of Transport and Communication, University of Zilina, 01026 Zilina, Slovakia; jaroslav.frnda@fpedas.uniza.sk; 3Department of Telecommunications, Faculty of Electrical Engineering and Computer Science, VSB-Technical University of Ostrava, 70800 Ostrava, Czech Republic; 4Department of Computer Science, Muslim Youth University, Islamabad 45710, Pakistan; nasim.abbas@myu.edu.pk

**Keywords:** 5G, SAS, CBRS, optimization, channel assignment, linear assignment problems, multiobjective

## Abstract

The paradigm of dynamic shared access aims to provide flexible spectrum usage. Recently, Federal Communications Commission (FCC) has proposed a new dynamic spectrum management framework for the sharing of a 3.5 GHz (3550–3700 MHz) federal band, called a citizen broadband radio service (CBRS) band, which is governed by spectrum access system (SAS). It is the responsibility of SAS to manage the set of CBRS-SAS users. The set of users are classified in three tiers: incumbent access (IA) users, primary access license (PAL) users and the general authorized access (GAA) users. In this article, dynamic channel assignment algorithm for PAL and GAA users is designed with the goal of maximizing the transmission rate and minimizing the total cost of GAA users accessing PAL reserved channels. We proposed a new mathematical model based on multi-objective optimization for the selection of PAL operators and idle PAL reserved channels allocation to GAA users considering the diversity of PAL reserved channels’ attributes and the diversification of GAA users’ business needs. The proposed model is estimated and validated on various performance metrics through extensive simulations and compared with existing algorithms such as Hungarian algorithm, auction algorithm and Gale–Shapley algorithm. The proposed model results indicate that overall transmission rate, net cost and data-rate per unit cost remain the same in comparison to the classical Hungarian method and auction algorithm. However, the improved model solves the resource allocation problem approximately up to four times faster with better load management, which validates the efficiency of our model.

## 1. Introduction

Over the last decade, traffic on telecommunication networks has experienced the exponential growth of wireless services due to diversity in internet-based applications. The astounding 1000-fold increase in data traffic in the coming decade demands fifth generation (5G) and beyond mobile communication systems, which will ensure high-speed and efficient data connectivity [1]. Recently, the Federal Communications Commission (FCC) has proposed to commercialize the spectrum used by the federal government to be made available for commercial internet service providers for licensed and unlicensed use [2]. FCC proposed an idea to share a citizen broadband radio service (CBRS) band on a commercial basis. The CBRS band is between 3550–3700 MHz held by the federal government for military use. Moreover, FCC also introduced three tier Spectrum Access System (SAS) architecture for the CBRS band, a unique framework for Dynamic Spectrum Management (DSM), which particularly addresses dynamic frequency assignment and interference management. The SAS is capable of assigning the CBRS band on a commercial basis to three types of users including incumbent access (IA) users also called federal military users having the highest priority to access the spectrum, primary access licensee (PAL) users are licensed users and general authorized access (GAA) users are unlicensed users having least priority [3,4,5]. In the CBRS architecture, a 70 MHz spectrum band is dedicated for PAL users having a channel of 10 MHz each. IA users are given full protection in their deployed areas while PAL users are protected from other PAL users as well as from GAA users. GAA users operate in GAA reserved channels. GAA users can also access the unused PAL reserved channels opportunistically. No interference protection is guaranteed for GAA users [6,7,8,9]. It is envisioned to adopt “use-it-or-share-it” policies to deter spectrum warehousing by sharing the PAL reserved channels to GAA users, but no details were provided to do so [10]. The methods for allocation of PAL reserved channels to GAA users are left open in the current release of the CBRS standard [11].

Currently, there has been comprehensive research and significant development in evolving techniques and mechanisms to adopt a shared spectrum access framework. The major challenges including CBRS radio spectrum allocation and management, interference management, inter-operator resource allocation, operational security and enforcement, standards for carrier aggregation and techniques for enabling harmonious coexistence of CBRS users with heterogeneous radio access technologies including Wireless Fidelity (WIFI) and Licenses Assisted Access (LAA) need to be addressed [12,13]. One of the main challenges in implementing shared spectrum access following CBRS architecture is the coexistence of PAL and GAA users in a census tract. A census tract is a geographical area, where PAL and GAA operators are allowed to provide the commercial services. Moreover, due to strict interference protection rules for high priority users, it is very difficult for low priority users to access the spectrum while high priority users are operating in that census tract. Furthermore, it imposes a major challenge to design an efficient channel assignment technique for PAL users as SAS is also responsible to assign associated channels to geographically associated PAL users. At the same time, the challenge to fairly allocate the resources to PAL and GAA users in multi-SAS environment in a census tract while limiting the interference and satisfying the rules proposed by FCC is still open and needs to be explored [14].

In this article, a mathematical model to allocate the idle PAL reserved channels to GAA users while satisfying QoS requirements and achieving a higher transmission rate at a minimum cost is proposed. In the proposed scenario, there are multiple PAL operators offering a variety of services with different network characteristics, i.e., spectrum width, delay, packet loss and spectrum price. GAA users request idle PAL reserved channels from PAL operators. As GAA users are least priority users, it is a challenging task to efficiently allocate the PAL reserved channels to GAA users at minimum cost as PAL operators paid for their license, and they have to protect PAL users from the interference caused by the addition of GAA users in the system.

This paper studies the diversity of available PAL resource attributes and GAA user requirements. The idle PAL reserved channels allocation to GAA users is modeled as a multi-objective optimization problem. In order to ensure the QoS to GAA users, the dual objective problem aims at maximizing the transmission rate and minimizing the cost to access idle PAL reserved channels. To solve this multi-objective idle channel allocation problem with different constraints, the optimization objectives with multiple constraints are simplified using a simplification method to obtain a simple polynomial time algorithm. Moreover, the classical Hungarian method is improved to solve the simplified optimization problem. The results of the improved method show that assignment of idle PAL reserved channels to GAA users satisfies the FCC rules defined including the QoS. The Improved Hungarian Method (IHM) assigns the idle PAL reserved channels to GAA users with maximum transmission rate and minimum cost. The improved Hungarian algorithm outperforms at higher network load in terms of execution efficiency. IHM allocates the idle PAL reserved channels to GAA users while satisfying FCC rules up to four times faster in comparison to the classical Hungarian method and up to three times in comparison with the auction algorithm and the Gale–Shapely stable matching algorithm at a higher network load.

Above all, the paper provides the following contributions:
1.A detailed mathematical model is defined for the channel allocation problem of CBRS-SAS architecture according to FCC rules to achieve higher transmission rate at minimum cost.2.We proposed an improved Hungarian method for solving the dual objective optimization problem that allocates the idle PAL reserved channels to GAA users while satisfying the rules proposed by FCC at minimum cost.3.We solved the defined optimization problem using the classical Hungarian algorithm, Gale–Shapley algorithm and auction Method.4.A detailed comparison of the four algorithms is given that shows that the improved Hungarian method is four times faster as compared to execution efficiency of the other three algorithms.

The rest of the article is organized as follows: Section 2 summarizes the related work. Section 3 presents the system model and detailed problem formulation. Problem solutions with the proposed solution are provided in Section 4. Results and performance evaluation are discussed in Section 5. Section 6 concludes our work.

## 2. Related Work

The problem of efficient utilization of the available spectrum has been the subject of intensive investigations for the last decade or so. The key focus is to create algorithms and techniques used to efficiently allocate the available resources to multi-tier users. In this context, many surveys have been published in recent years discussing the applicability of the techniques and protocols for coordination of licensed and unlicensed users in opportunistic spectrum access, cognitive radio networks (CRNs) and dynamic spectrum access [15,16,17]. The authors in [18] discussed various licensed spectrum sharing techniques in the context of their applicability by mobile network operators (MNOs). Authors in [19] presented the efficient radio access strategies for different spectrum bands opened for 5G and beyond. Three tier SAS based architecture for the 3.5 GHz CBRS band was investigated by the authors in [5] as a base paper in this domain. Idle channel allocation rules for multi-tier users were presented in [20]. The authors proposed a basic mathematical model to solve channel allocation to PAL and GAA users without considering the service requirements of GAA users. This paper extends this problem and considers the service requirements of GAA users as SAS does not guarantee GAA users for interference protection and providing required QoS. Static allocation of resources and the Max-Min fair technique were investigated in [14] to assign the available resources to GAA users. The authors consider the requirements proposed by FCC for the coexistence of different citizens’ broadband radio service devices CBSDs for PAL and GAA users as well as to ensure spectral efficiency.

The Wireless Innovation (WIN) forum investigated the spectrum coordination techniques for GAA users in the CBRS-SAS system based on graph coloring solutions discussed in [21,22,23]. The techniques proposed manage the GAA spectrum channels and interference between GAA users. In the recent paper [24], the authors proposed the centralized technique based on graph coloring with the add-edge algorithm and graph coloring with the reduced power algorithm for allocation of power and frequency among citizens broadband radio service devices (CBSDs). Graph theory-based resource allocation techniques are mostly applicable to simple and static environments. In the case of dynamic and distributed network architecture, where topology changes frequently, the efficiency is greatly compromised as every change in topology needs to recompile the complete allocation. Authors in [25] used the concepts of game theory and reinforcement learning to present the self-organized spectrum sharing technique and design of hybrid MAC (HMAC). Initially, Nash equilibrium in non-cooperative game is achieved to minimize interference and to share the spectrum among CBSDs. In another approach, the Markov decision process is used to model a Q-learning with a decision-making process technique by integrating different sources of interference. The authors in [26] proposed the super-radio formation algorithm using a carrier sensing mechanism from the WIFI domain to identify the set of radios coexisting in the same channel. An extended version presented in [8] by the same authors investigated SAS-assisted dynamic channel allocation for PAL and GAA users by introducing node-channel-pair conflict graphs to capture channel constraints, coexistence awareness, pairwise channel interference and spatially varying channel availability. Authors in [27] consider different user categories based on minimum acceptable throughput in which the proposed algorithm achieves the minimum requirements allocated to each category by assigning the resource blocks among all stakeholders. The objective in this paper was achieved by evaluating traffic requirements by calculating the shadowing effects and transmission path-loss. The dedicated field trials and hardware experiments to implement and evaluate CBRS-SAS architecture were discussed in [28,29]. Authors in [30] discussed the issues of detection of incumbent radar systems in the CBRS 3.5 GHz band. The resource allocation technique for shared spectrum access was investigated in [31] assuming terrestrial networks as incumbents and satellite networks. Authors in this paper proposed bandwidth and carrier allocation as well as joint power allocation schemes. In [32], the authors proposed a new concept of decentralized SAS based on the blockchain. The actual model proposed by FCC is based on centralized SAS so it is practically not possible to adopt the model. To implement SAS based CBRS architecture, we need to strictly follow the rules proposed by FCC as there will be high priority incumbent access users. It is the responsibility of SAS to protect them from low priority GAA users. Hence, centralized architecture is important to authorize GAA users. Authors in [33] presented privacy-preserving SAS framework in which blockchain technology is used to protect user’s privacy. GAA users share private information such as identity, spectrum usage and physical location. The authors use cryptographic mechanisms to protect this information. The mechanisms to share the spectrum with GAA users are left open. The authors in [34] used in-band-full-duplex (IBFD) techniques to protect Incumbent Access users in dynamic spectrum sharing based architectures from interference caused by low priority users. The authors designed the joint beamformers at mobile broadband networks (MBN) with the constraints on transmit power at MBN, beamformers at the radar system to alleviate interference and detection probability of a radar system. The authors discussed the interference mitigation techniques, but the resource allocation techniques were not considered. Sarkar et al. in [35] proposed an enhanced environmental sensing capability sensor based on deep learning to estimate spectral occupancy and achieved high detection accuracy.

The authors in [36] discussed the spectrum allocation in the context of heterogeneous radio access technologies. The problem is modeled as a dual-optimization problem in which secondary users are accessing the idle channels from primary networks including Cellular, Wi-Fi and the Worldwide Interoperability for Microwave Access (Wimax) networks. The problem is then solved using the Hungarian Method and a non-sorting genetic algorithm version II (NSGA-II) algorithm, and the results are compared with the multi-objective artificial bee colony algorithm. However, the improved NSGA-II method proposed takes a lot of time to allocate the users as it depends on the number of iterations to reach the optimal solution. Hence, it is better to solve the multi-objective optimization problem using improved linear assignment problems to get the best results.

As mentioned above, notable research has been made in spectrum allocation and network selection; however, there is still a need for a solution for dynamic spectrum allocation in a multi-tier environment as game theory based spectrum allocation techniques are used when the spectrum users interact directly, in order to analyze the competition and cooperation. In the case of a CBRS-SAS based system, SAS is responsible to authenticate and allocate the channels to all tiers. Spectrum trading techniques and models are suitable in a scenario of cognitive radio networks (CRN) where primary and secondary users are leased. Moreover, the above-mentioned research articles mostly focus on traditional spectrum allocation issues, while ignoring the QoS and service requirements of GAA users. Furthermore, most channel allocation techniques available in the literature do not satisfy the rules proposed by the FCC for resource allocation or a lack of providing required QoS. In this article, taking into account the requirements of PAL operators and GAA users, more pragmatic application requirements are considered. Thus, this paper comprehensively targets the allocation of spectrum resources to GAA users with guaranteed QoS at minimum cost while satisfying FCC rules.

## 3. System Model

### 3.1. Network Model

A 3.5 GHz spectrum range was reserved for the use of US Navy radar systems. FCC identifies the under-utilization of this spectrum band and released a 150 MHz broadcast band for commercial wireless services. The 3550–3700 MHz band was open to mobile network operators to provide services with and without license with a feature of dynamic spectrum access system SAS. Hence, three-tier layered architecture has been proposed for a CBRS 3.5GHz band to accommodate the incumbent access IA users including US Navy radars and new commercial users including licensed and unlicensed users. The layered architecture is shown in Figure 1. CBRS-SAS network architecture is represented in Figure 2. A CBRS-SAS system is comprised of the following elements:1.Spectrum Access System (SAS);2.Environmental Sensing Capability (ESC);3.Citizen Broadband Radio Service Device (CBSD);4.Domain Proxy (DP);5.Network Management System (NMS);6.FCC External Databases;7.End Devices.

SAS repository is responsible for allocating frequency channels to three types of users. It protects the upper tier users from the interference caused by lower tier users and maintains the QoS for PAL users. It also assigns transmission power to citizen radio service devices CBSDs.

An eNodeB capable of supporting a CBRS band is called CBSD. To register a CBSD, SAS uses external FCC data bases to ascertain the transmit power and number of assigned channels to a grant spectrum. SAS specifies the assigned frequency spectrum and required effective isotropic radiated power (EIRP) at which the CBSD can transmit.

Environment sensing capability (ESC) detects the incumbent’s activity and tells SAS to vacate the channels from lower tiers. Domain Proxy (DP) is deployed to manage the proxy functions and CBSD aggregation for large scale networks, and it can also be integrated with a network management system (NMS) or element management systems (EMS). PAL and GAA users use end Internet of Things (IoT) enabled devices that support a CBRS frequency band in connection with a CBSD. In this article, multiple GAA users are assigned channels from PAL reserved channels with overlapping coverage in a census tract. The secondary users are randomly deployed and assigned channels from GAA reserved channels as well as from PAL reserved channels at reasonable cost while limiting the overall interference threshold.

Let the number of incumbent users be *N*. The incumbent user’s set is given by:$$I=\{I_{1},I_{2},\ldots \ldots ,I_{i}\}$$
The set of PAL users is given by:$$P=\{P_{1},P_{2},\ldots \ldots ,P_{p}\}$$
and GAA user’s set is comprised of
$$G=\{G_{1},G_{2},\ldots \ldots ,G_{g}\}$$
Let *S* represent the set of frequency spectrum available in a census tract, Sp denotes the set of PAL reserved channels in a census tract such that Sp⊆S and PAL reserved channels set is represented as:$$Sp=\{S_{1},S_{2},\ldots \ldots ,S_{m}\}$$
The set of GAA reserved channels is shown as:$$Sg=\{S_{m+1},S_{m+2},\ldots \ldots ,S_{n}\}$$
and the set of all channels in a census tract is given as
$$S=\{S_{1},S_{2},\ldots ,S_{m},S_{m+1},S_{m+2},\ldots \ldots ,S_{n}\}$$

Moreover, the behavior of each IA and PAL users is independent. It is modeled as a Poisson process of switch 2 state, i.e., idle or busy. State 0 indicates the idle state, and state 1 represents the busy state. In state 0, no IA or PAL user is using the spectrum resources. In state 1, IA or PAL users are active in a particular channel that cannot be assigned to GAA users.

GAA user’s transmission rate is related to its transmission power, spectrum bandwidth and SNR. According to Shannon’s theorem, GAA user $g^{\prime }s$ transmission rate can be expressed as:(1)$$t_{g}=b_{g}^{s_{m}}y_{g}^{s_{m}}log_{2}\left(1+\frac{S_{g}^{s_{m}}}{N_{g}^{s_{m}}}\right)$$
where $b_{g}^{s_{m}}$ indicates the total bandwidth assigned to GAA user *g*, $y_{g}^{s_{m}}$ represents the probability of a GAA user to access channel $s_{m}$. 1 indicates that the spectrum is allocated, and 0 shows that the spectrum is not occupied. The ratio of $S_{g}^{s_{m}}$ and $N_{g}^{s_{m}}$ shows the signal-to-noise ratio of GAA user *g* occupying spectrum $s_{m}$. Finally, Table 1 summarizes the notations list used.

### 3.2. Problem Formulation

This article focuses on the allocation of idle PAL reserved spectrum resources to GAA users in the presence of PAL users while protecting them from any noticeable interference with the goal of achieving a higher transmission rate at a reasonable cost. Different GAA users have diverse QoS requirements for different applications such as video, voice and file transfer. These applications have different delay, bandwidth and packet loss rate requirements. Therefore, it is necessary to allocate idle PAL reserved channels to GAA users based on PAL network attributes and service requirements of GAA users. Moreover, when a GAA user accesses the PAL reserved channel, it will cause an interference to PAL users, due to which the quality of PAL users could be seriously affected. SAS must define an interference threshold to limit the access to GAA users to protect the QoS of PAL users. The SAS ensures to provide minimum inter-GAA interference by frequency reuse between GAA users, whose inter-CBSD distances are greater. For this, let us first define
(2)$${D.f}_{hg}^{S_{m}}=\left\{\begin{array}{cc} 1, & \mathrm{if}\;y_{h}^{S_{m}}and\;y_{g}^{S_{m}}=1\\ 0, & \mathrm{otherwise} \end{array}\right.$$
${D.f}_{hg}^{S_{m}}$ is the distance factor that is equal to 1 if both GAA users *h* and *g* are using the same channel $S_{m}$ and ${D.f}_{hg}^{S_{m}}$ = 0 if both users are using different channels. Now, the inter-CBSD distance can be defined as discussed in [20]:(3)$$D_{icbsd}={D.f}_{hg}^{S_{m}}{\left(D_{hg}\right)}^{-\beta }$$

$D_{hg}$ represents the distance between *h* and *g* GAA users, and β is the positive constant that regulates the distance cost. If β is smaller, then costs added to the overall distance will be large and vice versa.

Considering these constraints, the channel allocation problem is modeled as a dual-objective optimization model. The objective of this model is to maximize the GAA user’s transmission rate while minimizing the cost.

The cumulative transmission rate of all GAA users based on Equation (Equation 1) can be represented as
(4)$$T=\sum_{s_{m}=1}^{m}\sum_{g=1}^{g}b_{g}^{s_{m}}y_{g}^{s_{m}}log_{2}\left(1+\frac{S_{g}^{s_{m}}}{N_{g}^{s_{m}}}\right)$$

The problem modeled in this article for resource allocation is defined as a nonlinear constraint 0–1 integer programming problem i.e.,
(5)$$max\;\;\left(T\right)=max\sum_{s_{m}=1}^{m}\sum_{g=1}^{g}b_{g}^{s_{m}}y_{g}^{s_{m}}log_{2}\left(1+\frac{S_{g}^{s_{m}}}{N_{g}^{s_{m}}}\right)$$
(6)$$min\;\;\left(C\right)=min\sum_{g=1}^{g}\sum_{s_{m}=1}^{m}c_{g}^{s_{m}}y_{g}^{s_{m}}$$
subject to the constraints:(7)$$I_{i}^{S_{n}}\sum_{g\in G}y_{g}^{S_{m}}=0\;,\;i\in I,\;S_{m}\in S$$
(8)$$\sum_{p\in P}x_{p}^{S_{m}}.\sum_{g\in G}y_{g}^{S_{m}}=0\;,\;S_{m}\in Sp$$
(9)$$t_{g}^{S_{m}}.y_{g}^{S_{m}}\leq t^{S_{m}}\;,\;g\in G,\;S_{m}\in S$$
(10)$$c_{g}^{S_{m}}.y_{g}^{S_{m}}\leq c^{S_{m}}\;,\;g\in G,\;S_{m}\in S$$
(11)$$d_{g}^{S_{m}}.y_{g}^{S_{m}}\leq d^{S_{m}}\;,\;g\in G,\;S_{m}\in S$$
(12)$$l_{g}^{S_{m}}.y_{g}^{S_{m}}\leq l^{S_{m}}\;,\;g\in G,\;S_{m}\in S$$
(13)$$\sum_{g\in G}y_{g}^{S_{m}}.Q_{g}^{S_{m}}\leq Q^{S_{m}}\;,\;g\in G,\;S_{m}\in Sp$$
(14)$$\sum_{s_{m}=1}^{m}y_{g}^{S_{m}}\leq 1\;,\;\forall \;g=1,2,\ldots ,g$$
(15)$$\sum_{g=1}^{m}y_{g}^{S_{m}}\leq 1\;,\;\forall \;s_{m}=1,2,\ldots ,m$$
(16)$$y_{g}^{S_{m}}=\left\{0,1\right\}$$
maxT represents the objective function in Equation (Equation 5), *T* represents the cumulative transmission rate achieved by all GAA users, *g* represents the GAA user and $S_{m}$ refers to the available idle spectrum in PAL reserved channels. $b_{g}^{S_{m}}$ represents the bandwidth obtained by GAA user *g* in channel $S_{m}$, and $y_{g}^{S_{m}}$ is the probability of GAA user *g* occupying the spectrum resources $S_{m}$. Equation (Equation 6) represents the objective function to minimize the cost of GAA users to access PAL reserved channels. $c_{g}^{S_{m}}$ indicates the cost of GAA user *g* for accessing PAL reserved channel $S_{m}$.

Equation (Equation 7) represents that channel $S_{m}$ will not be allocated to GAA user *g* in the presence of IA user *i*. Equation (Equation 8) shows that channel $S_{m}$ will be allocated to either PAL user *p* or GAA user *g*. Equations (Equation 9)–(Equation 12) indicate that the constraints must meet the requirements of transmission rate, cost, delay and packet loss, respectively. Equation (Equation 13) represents that the net interference caused due to allocation of channel $S_{m}$ to GAA user *g* remains under the interference threshold limit $Q^{S_{m}}$. Equation (Equation 14) represents that the idle PAL reserved channel $S_{m}$ can be allocated to any GAA user and a channel $S_{m}$ cannot be assigned to more than one GAA user at the same time. Equation (Equation 15) denotes that two GAA users cannot be allocated to the same channel, and Equation (Equation 16) ensures that the variable $y_{g}^{S_{m}}$ is the binary variable that shows the probability of whether or not the GAA user occupied the channel $S_{m}$.

### 3.3. Problem Complexity Analysis

As discussed in Section 3.2, the number of PAL reserved channels and GAA users are represented by *p* and *g*, respectively. The channel allocation to GAA users from PAL reserved channel is shown in Table 2 below. In this table, the probability of GAA users occupying idle PAL reserved channels is represented. If a channel is occupied by a GAA user, then it is represented by 1, otherwise 0. There is a one-to-one relationship between GAA users and PAL reserved channels as discussed in Section 3.1 in which a GAA user acquires one idle channel from the set and that channel cannot be allocated to another GAA user. Accordingly, a maximum one non-zero element can be placed in the same row or column. Thus, the channel allocation problem in this article is a 0–1 programming problem. In Section 3.2, the detailed mathematical model of a channel allocation problem is discussed in which there are two optimization objectives subject to multiple constraints. This problem is a nonlinear multi-objective optimization problem and considered as an NP-Hard problem [37]. Hence, computational complexity is directly proportional to the scale of the problem; if the scale of the problem expands, the computation complexity will be increased with factorial level [38].

## 4. Problem Simplification and Optimization

The analysis in Section 3.3 concludes that the defined spectrum allocation problem is an NP hard problem. This article simplifies the mathematical model defined in Section 3.2 to solve this problem. The problem is also solved using the classical Hungarian algorithm, auction algorithm and stable matching algorithm. Finally, the applied methods are compared to evaluate the performance of each method.

### 4.1. Simplification Process

The simplification process proposed in this article is the improved version of the classical Hungarian algorithm. The Hungarian algorithm executes the optimal assignment problems as combinatorial problems to solve the *n*×*n* task allocation problem in $O(n^{3})$ time [39]. The algorithm has two main steps. The first step is to modify the matrix to add interference value constraints to the test assignment of the Hungarian algorithm. In the next step, the coefficient matrix is checked regarding whether it is a square matrix or not. If the matrix is not square, then add virtual GAA users or virtual channels to make it a square matrix. In the final step, each value of the matrix is subtracted from the maximum value of the matrix to remodel the maximization problem into a minimization problem. After applying aforesaid preprocessing steps, the improved Hungarian algorithm is applied to allocate PAL idle reserved channels to GAA users. The detailed steps and time complexity of the simplified method are discussed below.

#### 4.1.1. Simplified Method Steps

As discussed in Section 3, it can now be concluded that the spectrum allocation with the dual objective optimization problem, defined in this article to achieve a higher transmission rate and to minimize the cost to access an idle spectrum, is a non-standard 0–1 programming problem.

Table 3 represents the bandwidth efficiency matrix, which indicates the net bandwidth achieved when idle channels are allocated to GAA users. The transmission rate is calculated using Equation (Equation 1) for corresponding idle PAL reserved channels.

Table 4 shows the cost–benefit matrix that represents the net cost to access the idle spectrum. The net cost for accessing each PAL reserved channel is calculated using Equation (Equation 6).

In the spectrum allocation problem, the interference limitation and QoS requirements of GAA users are also considered. Therefore, the classical Hungarian algorithm cannot solve this problem.

In order to reduce the computational complexity and to simplify the mathematical model, the following steps are used: Initially, a new efficiency factor was introduced called the relative transmission rate–cost factor α that converts the multi-objective optimization problem of maximizing transmission rate and minimizing cost into a single objective function. In the next step, constraints are simplified to convert the spectrum allocation problem to a standard 0–1 integer programming problem, and this problem is solved using a Hungarian algorithm. The main steps of Hungarian algorithm are as follows:

#### 4.1.2. Transforming Objective Function

The defined objective functions in Equations (Equation 5) and (Equation 6) aim to achieve a higher transmission rate at minimum cost. In order to convert the multi-objective function into a single objective function, a relative transmission rate–cost factor α is introduced, which is defined as
(17)$$\alpha_{g}^{S_{m}}=\frac{r_{g}^{s_{m}}}{c_{g}^{s_{m}}}$$
where $r_{g}^{S_{m}}$ and $c_{g}^{S_{m}}$ indicate the transmission rate and cost, respectively. $\alpha_{g}^{S_{m}}$ denotes the the transmission rate obtained by GAA user *g* while accessing channel $S_{m}$ at cost *c*. Now, after defining α, the objective Equations (Equation 5) and (Equation 6) in Section 3.2 are redefined as follows:(18)$$Max\;\;\alpha \;=\sum_{g\in G}\sum_{S_{m}\in S_{p}}\alpha_{g}^{S_{m}}y_{g}^{S_{m}}$$
Therefore, the transmission rate efficiency matrix and the cost efficiency matrix in Table 3 and Table 4, respectively, can be redefined in the form of a relative transmission rate–cost efficiency matrix defined in Table 5.

#### 4.1.3. Simplification of Constraints

Assume that there are no IA users and channel $s_{m}$ is idle and available to be allocated to GAA users. Hence, Equations (Equation 7) and (Equation 8) are eliminated. Equation (Equation 9) ensures that the transmission rate of PAL reserved idle channel that is allocated to GAA user *g* must be greater than the requirement of GAA users. Hence, the transmission rate and cost factor α will be set to 0 if transmission rate is less than the requirement. Similarly, in Equations (Equation 10)–(Equation 12), if the constraints that do not meet the requirements rather than factor α will be set to 0. Equation (Equation 13) represents that the interference experienced by PAL users by assigning channels to GAA users must be within PAL threshold limits. Interference in the PAL network is gradually increased whenever GAA users access the PAL network so that this constraint cannot be ignored using the same method, as this is the dynamic process. Therefore, an improved Hungarian algorithm is proposed in which Equation (Equation 13) is considered in the step of allocating 0 elements.

#### 4.1.4. Normalized Preprocessing

Since GAA users accessing the PAL reserved channels are not equal to the number of idle channels available (g≠m), the channel assignment problem here is an unbalanced problem. Normalized preprocessing is required before applying the improved Hungarian algorithm. The details of preprocessing are given as follows:1.When idle channels available are equal to the number of GAA users requesting the channels (g=m), there is no need to deal.2.When GAA users are greater than the available idle channels (g>m), add (g−m) virtual resources and mark their relative transmission rate cost factor α as 0.3.When GAA users are less than available idle channels (g<m), then mark 0 for their relative transmission rate cost factor α of the row where the GAA user is located.

Simultaneously, the maximization of relative transmission rate–cost factor α is converted to a minimization problem i.e., determine the $\alpha_{max}$ in the efficiency matrix in Table 5 and then subtract each element of that matrix from the maximum value $\alpha_{max}$ to obtain a new efficiency factor γ that is expressed as $\gamma_{gm}$ = $\alpha_{max}-$$\alpha_{gm}$ as shown in Table 6.

### 4.2. Improved Hungarian Algorithm

The Hungarian algorithm is a polynomial time algorithm used to solve allocation problems. The algorithm has high efficiency and simple steps. However, the classical Hungarian algorithm does not consider interference constraints. The algorithm was improved to integrate the interference value with a trial assignment step to keep the interference value below the PAL network interference threshold limit. The flow of the classical Hungarian algorithm and the key step “transformation of efficiency matrix” is improved to enhance the efficiency of the standard Hungarian algorithm. The main steps of improved Hungarian algorithms are:

#### 4.2.1. Transformation of Efficiency Matrix

In the traditional Hungarian algorithm, each element of the row and column is subtracted from the row minima and column minima to obtain “0” elements. In the improved version, first determine the minimum number of each row and each column. If the row minima is less than the column minima, start to subtract the row elements from the row minima; otherwise, subtract the column elements from the column minima. This technique increases the number of “0” elements and the probability of successful trial allocation, and improves overall execution efficiency.

#### 4.2.2. Trial Assignment

If the row or column contains only one “0” element after subtracting the row and column from the corresponding minima, it is called independent “0”. In this step, an independent “0” element is determined for each row and finds the corresponding PAL channel to which it found. If the interference value does not surpass the PAL network interference threshold limit after the addition of a GAA user corresponding to the particular row, then assign the “0” element to a GAA user and update the PAL network interference threshold values. Following the same process, find the independent “0” element in each column in turn. Use the circular search algorithm and trial assignment to find the independent “0” element for each row and column until all independent “0” elements are allocated. The next step is to assign non-independent 0 elements. There are still some 0’s left behind after independent 0 assignment. Find the left 0 elements in each row and column. Calculate the minimum number of 0 elements in each row and column; if there is an equal number of 0 elements in both rows and columns, then randomly choose from the row or column with the least number of 0 elements. Determine the PAL channels to which the 0 elements belong and repeat the step of interference value processing. If the unallocated 0 element meets the GAA service requirements, then assign the 0 element to a particular GAA user and perform the assignment labeling. The final step is to repeat a circular or loop search algorithm and trial assignment to find the “0” element for each row and column until no 0 elements are left. Find the total number (total) of ‘0’ elements assigned. If total and efficiency matrix dimensions (n) are equal, then it is an optimal solution. If total is not equal to *n*, then proceed to Section 4.2.3.

#### 4.2.3. Cover all Zeros with a Minimum Number of Lines

Cover all 0 elements in the resultant matrix using a minimum number of horizontal and vertical straight lines denoted by *l*; if *n* lines are required (l=n), then repeat Section 4.2.2 for reallocation. If less than *n* number of lines are required (l<n), continue with Section 4.2.4.

#### 4.2.4. Create Additional ‘0’ Elements

Calculate the minimum value in the resultant matrix that is not covered by a line in Section 4.2.3 denoted by *u*. Subtract *u* from all uncovered elements in the unmarked line. Add *u* to all unmarked elements in a column that is unmarked and proceed to Section 4.2.2.

Applying the improved Hungarian algorithm on the multi-objective optimization problems defined in Equations (Equation 5) and (Equation 6) in Section 3.2 is simplified as:(19)$$max\;\;\alpha \;=max\sum_{g\in G}\sum_{S_{m}\in S_{p}}\alpha_{g}^{S_{m}}y_{g}^{S_{m}}$$
subject to
(20)$$\sum_{g\in G}y_{g}^{S_{m}}\leq 1\;,\;S_{m}\in S_{p}$$
(21)$$\sum_{S_{m}\in S_{p}}y_{g}^{S_{m}}\leq 1\;,\;g\in G$$
(22)$$y_{g}^{S_{m}}=\left\{0,1\right\}$$

#### 4.2.5. Time Complexity Analysis

In Section 4.1.1, the dual optimization spectrum allocation problem is transformed into a single objective optimization problem. The problem is then transformed into a 0–1 programming assignment problem by eliminating constraints using a simplification process. Moreover, the improved Hungarian algorithm is used to solve this problem. Consider that the number of GAA users accessing the PAL reserved idle channels and the number of idle channels available are equal after standardization. It is observed from Section 4.2.2 of the trial assignment step of the improved Hungarian algorithm that the time for independent 0 assignment is the significant execution time. The time complexity of row independent 0 elements assignment is $O(n^{2})$, as, for the assignment of row independent 0, each element of each row must be searched. In the same way, for assignment of column independent 0 elements, the time complexity is defined as $O(n^{2})$. Therefore, the time complexity for Section 4.2.2 is $O(n^{2})$. In the third Section 4.2.3 of the improved Hungarian algorithm, it is checked whether idle channels are allocated to GAA users or not. If allocation is not yet done, then it proceeds to the next Section 4.2.4 to increase 0 elements by transforming the efficiency matrix and jump back to Section 4.2.2 for trial assignment. In the worst scenario, Section 4.2.4 is repeated *n* times to increase 0 elements; therefore, the Section 4.2.2 will also be performed *n* times cyclically. Hence, the time complexity is $O(n^{3})$ for the simplification method. Accordingly, it is concluded that the improved Hungarian algorithm is an efficient algorithm to solve idle resource allocation in a polynomial time.

### 4.3. Gale–Shapley Stable Matching Algorithm

The optimization problem defined in Section 3.2 has exponentially increasing computational complexity over the network size. This optimization problem can be solved by modeling the problem as a student admission matching problem defined in [40]. The GAA users and available idle channels can be considered as matching parties, respectively. The factors like transmission rate and cost are used to build the preferences each party has. This problem then will be solved using a distributed algorithm i.e., a Gale–Shapley stable matching algorithm. In the student admission matching algorithm, there is a student request for admission to a particular college. The college processes the request and decides whether to admit or not. The student ranks and sets the preference for the college based on different factors, and colleges rank the students based on their submitted educational profile.

Similarly, the optimization problem is mapped to this scenario in which SAS sets up preference lists for GAA users and service providers, respectively. A GAA user having multiple idle PAL reserved channels available has to select a channel based on receiving data rate and cost. The service provider has its own preferences to allocate channels with the highest cost. There are limited channels that are available to be assigned and if the users requesting the channels are greater than available channels, then SAS has to decide to assign the channels to GAA users based on preference of both parties and reject the rest. After setting the preference list, SAS checks the requests of GAA users and service provider cost preference and assigns the most preferred channel to GAA user and rejects the rest. The process goes on until all the users are assigned the channels or there are no more idle channels available. The output of this algorithm is a stable matching that does not assign any blocking pair.

The key steps of the Gale–Shapley Algorithm are as follows.

1.In the first step, SAS sets up the preference list of GAA users and available idle channels. In the first iteration request, every GAA user is sent to the preferred available channel. Each channel responds with “maybe” to the suitor that is the most preferred and no to all other users. The available channel is then provisionally engaged to the GAA user that is most preferred.2.In the following iterations, each GAA user that is still unassigned sends a request to the preferred channel to whom it has not been sent irrespective of whether the channel is allocated or not. Each channel that receives the request accepts the request if the channel is not assigned yet or it is preferred over an existing provisional user by replying “maybe”. Consequently, the provisional user becomes un-engaged by accepting the new user to the idle channel. This provisional assignment of channels conserves the right of the previously assigned channel to trade up.3.The iterations will be repeated until all GAA users are assigned.

The time of complexity of this algorithm is $O(n^{2})$, where *n* shows the number of channels and number of GAA users. The runtime behaves linearly as the size of preference lists are also proportional to $\left(n^{2}\right)$. The algorithm ensures that each user is allocated and gives a stable match instead of the best preferred match.

### 4.4. Auction Method

The auction algorithm was proposed to solve the classical transportation problem as an assignment problem [41]. The algorithm practically and theoretically solves the assignment problem in a better way as compared to its competitor algorithms. The algorithm is based on a real bidding system, where a bidder increases the bid of a preferred product by some bidding increment to get the product. The classical assignment problem deals with *n*x*n* persons and products to be assigned, where each person is assigned to its preferred product. In this scenario, the objective of the problem is to allocate the preferred product to the person who is the best bidder. To solve this problem, the auction algorithm uses the concept of bidding and cost equilibrium. Supposing that there are *n* persons, a person *x* has his own interests. Let us assume that there are *n* products and the cost to get the preferred product *y* is $c_{p}$. Person *x* must pay the cost $c_{p}$ to get the product. A factor $f_{xy}$ is introduced to determine the net value of a person for its preferred product. Mathematically, it can be written as value = $f_{xy}$ - $c_{p}$. To allocate each product to the person, the particular factor value must be at a maximum. Hence, all the persons *x* allotted to products *y* having a maximum factor value meet the net cost equilibrium, and persons are set to satisfied with the products assigned.

The key steps of auction algorithm are:1.Initialization: All persons are randomly allocated to the products, and the cost of each product $c_{p}$ is set to 0.2.Determine minimum bidding amount: Determine the price of minimum bid *b* that is set to 1/(n−1), where *n* is the number of persons or products that participate in the auction process. The value of *b* is set to 1/(n−1) because the auction algorithm with this value does not converge, and a lesser value is selected.3.Select a random unsatisfied person.4.Product exchange: In this step, the product of an unsatisfied person is replaced with the person allocated to product $p_{k}$ in step 1. The updated cost of a bidding product is represented by $newc_{p}$ = $oldc_{p}$ + $inc_{x}$ + value, where inc is the increment to raise the cost of product $p_{k}$ such that the product remains the preferred product for person *x*. Increment can be represented by $inc_{x}$ = $bv_{x}$ – $wv_{x}$, where $bv_{x}$ is the best value of a product for person *x*, and $wv_{x}$ is the worst value of a product for person *x*.5.Updating profits: Subtract the new bidding cost $newc_{p}$ from the profit of allocating product *k* to each person *x* and update the profits of product $p_{k}$6.Repeat process: Repeat the step 3 by selecting a new unsatisfied person and complete the allocation until every person is satisfied with the allocated product and cost equilibrium is achieved.

The auction algorithm will converge faster if step 3 is replaced with the step of selecting the most unsatisfied person instead of randomly selecting an unsatisfied person. In comparison with the classical Hungarian algorithm, the auction algorithm is slower as it iterates through a different number of iterations for each product because of a random selection of unsatisfied persons. The cost for each assignment in all test runs remains the same and is consistent for both algorithms. In case the auction algorithm is selected while selecting the most unsatisfied person at step 3, then it will converge faster in comparison with the Hungarian algorithm.

## 5. Simulation and Results

### 5.1. Simulation Setup

In this section, the performance of the classical Hungarian, improved Hungarian, Gale–Shapley and auction algorithms are evaluated using Matlab simulations. In our simulations, we consider that there are four PAL operators and three GAA operators. Each PAL operator has a different number of PAL users randomly deployed with a set of idle spectrum available. It is assumed that a GAA user can request a maximum of two channels from PAL operators. The SNR received at distance $d_{cbsd}$ is deifned in [20] is represented by:(23)$$SNR\left(d_{cbsd}\right)=\frac{MaxP^{T}}{\delta^{2}}{\left(\frac{d}{d_{0}}\right)}^{\eta }$$
where $MaxP^{T}$ represents the maximum transmit power of PAL operator, η is set to 4 and $d_{0}$ is set to 1. The factor $\frac{MaxP^{T}}{\delta^{2}}$ = 40 dB. The transmit power is considered as the same for all PAL operator CBSDs and GAA users. The interference at the boundary of PAL operator CBSDs is calculated by using a path loss model defined in [42]. β is set to 2 and interference threshold in our simulations is set at −25 dB. Table 7 represents the characteristics of each PAL network operator that is offering QoS services to GAA users with the details of available bandwidth, cost demand, QoS parameters and interference threshold.

GAA users can select one of three services from video, voice or data. Each GAA user has its own set of service requirements that may have different costs, bandwidth and QoS parameters. GAA users requirements for each service is shown in Table 8.

To evaluate the performance of proposed algorithms, the number of available idle PAL reserved channels are assumed to be in the range of 1–500 and, to evaluate the performance of algorithms, GAA users’ load on network is increased to 5–500. PAL operators accept the GAA users’ requests until the PAL network reaches its interference threshold limit.

### 5.2. Performance Evaluation

In this article, efficient channel assignment to achieve maximum transmission rate at minimum cost using the improved Hungarian algorithm, the classical Hungarian algorithm, auction algorithm and Gale–Shapley stable matching is evaluated and compared. To minimize the randomness from experiment results and get the stable patterns, each experiment conducted was repeated 50 times to obtain the mean value.

The cumulative distributed function (CDF) of channel reuse distance for all the channels allocated to GAA users is depicted in Figure 3 for the scenario when a GAA user is provided with one channel and two channels. It is evident from the graph that, for a given scenario, when a GAA user requires a single channel, the probability of channel reuse distance of 100 units is 0.2 approximately and interference remains under the threshold limit, while, in a scenario where a GAA user requires two channels, the probability of channel reuse distance of 100 units is around 0.7. When a GAA user is provided with two channels, SAS allocates the same channel to two different users so that the inter-CBSD distance is decreased, which is the reason for increased interference experienced by GAA users.

Figure 4 represents the comparison of aggregate interference experienced by GAA users accessing PAL idle reserved channels when a GAA user requires a single channel and when a GAA user requires two channels. It is clear from the result that the overall aggregate interference in the case when a GAA user requires two channels is increased because, in order to fulfill the channel requirement of GAA users, the same channel is allocated to more than one user. In a scenario when a GAA user requires a single channel, interference remains under the threshold limit. Hence, interference protection is guaranteed to GAA users.

Figure 5 and Figure 6 show the PAL reserved idle channels’ allocation to the average number of GAA users when the GAA users’ channel requirements are 1 and 2, respectively. It is clear from the results that, when a GAA user needs two channels, the minimum average number of GAA users assigned to each idle channel is almost double the scenario in which a GAA user requires a single channel.

GAA users may ask for multiple channels, and this information will be provided to SAS. However, according to FCC rules, it is not guaranteed to always serve GAA users as per their QoS requirements, as GAA users are the least priority, and the SAS has to protect IA and PAL users from the interference caused by GAA users. When GAA users require more than one channel to meet their QoS requirements, the available PAL reserved channels will be allocated to more users that will cause more interference, as is evident from Figure 4. Figure 5 and Figure 6 show that an available PAL reserved channel can be allocated to how many users when GAA users’ channel requirement is 1 and 2, respectively. We considered a scenario in which there are six idle channels available and SAS has to assign these channels to GAA users as per their QoS requirements. Figure 5 shows the result of an experimental setup, where, on average, a single channel is required by GAA to meet their QoS requirements. In this experiment, the average number of users allocated to different channels ranges from 0.2 to 3. As the QoS requirements of GAA users are enhanced and the demand of channels is doubled to meet the QoS requirements, the channel allocation is also doubled as depicted in Figure 6. This consequently results in increased interference as is evident from Figure 4, where the interference exceeds way beyond the interference threshold defined in CBRS Alliance technical specifications of [5], i.e., –25 dB. This leads to the conclusion that, in order to accommodate the GAA users in the system, the average number of channels required by GAA users is an important consideration. As GAA users are least priority users, SAS will not accommodate them if the interference for PAL users exceeds the interference threshold limit. Thus, to accommodate GAA users in PAL reserved idle channels, SAS considers the GAA users with a channel requirement equal to 1 such that the net interference remains under the threshold limit and then maximum GAA users can be accommodated. This will ensure an acceptable accommodation of GAA users in the PAL reserved channels and also the efficient use of network resources.

Figure 7 represents the net data rate achieved. The Hungarian algorithm, improved Hungarian algorithm, and auction algorithm give the highest net data rate achieved by all GAA users accessing PAL reserved channels. Gale–Shapley gives the optimum and stable results for all the channels and network load as it finds the stable match instead of finding the maximum value. Similarly, Figure 8 shows that the Hungarian algorithm, the improved Hungarian algorithm and the auction algorithm are giving the minimum net cost offered to PAL operators for accessing PAL reserved channels, while the Gale–Shapley algorithm is again giving the stable match. In Figure 9, the data rate per unit cost is represented. Gale–Shapley is giving very less data rate per unit cost. The auction algorithm, classical Hungarian method and the improved Hungarian method are giving the highest data rate per unit cost.

As discussed in Section 4.3, the Gale–Shapley stable matching algorithm determines the stable pairing of user and idle PAL reserved channels according to the weight assigned to each user. If the position of a GAA user changes, it shows that there is a new GAA user who is considered. The algorithm maintains the preference table that saves the interest of both the GAA users and PAL operators having idle PAL reserved channels. Any matching that is made between GAA users and idle channels in a set is called an allocation of stable pairs. Stable matching is an iterative process, where the temporary pairing is updated in each iteration until the stable pairs are determined in the last round of iterations and finally the channels are assigned to the GAA users. However, the Hungarian algorithm, auction algorithm and improved Hungarian algorithm find the maximum value according to the requirements of each party i.e., in our scenario, these algorithms select the PAL idle reserved channel with maximum transmission rate at minimum cost and the Gale–Shapley stable matching algorithm finds the stable pairs, which may not be yielding optimal data rate per unit cost. In Figure 7, Figure 8 and Figure 9, it is evident from the graphs that the Hungarian algorithm, auction algorithm and the improved Hungarian algorithm result in a maximum data rate at minimum cost which yield optimal data rate per unit cost. As these three algorithms converge to the same values, which corroborates our results, thus the ultimate deciding factor for SAS to choose the algorithm to allocate the idle channels to its users depends on the computational complexity of the algorithm.

Figure 10 depicts the execution time of four algorithms to assign channels to GAA users. The improved Hungarian algorithm shows the best results when the users load over the network is increased. The improved Hungarian algorithm assigns 500 available channels to GAA users in 0.2 s that is approximately four times faster than the classical Hungarian algorithm. The proposed algorithm is approximately 2.5 times faster than the Gale–Shapley and auction algorithm. The classical Hungarian algorithm is performing the worst in comparison. The time taken by algorithms to assign channels is gradually increasing as the load on system is increased. The improved Hungarian method is taking up to four times less time at the higher network load as the flow of the classical Hungarian method is improved as discussed in Section 4.2.

The compiled simulation results show that the improved Hungarian method is giving the highest net data rate at minimum net cost in less time with approximately four times faster execution efficiency as compared to the classical Hungarian method. Hence, its efficiency is much better to allocate the idle PAL reserved channels to GAA users. The classical Hungarian method, auction method and improved Hungarian method give the highest values of data rate and minimum values of cost while the Gale–Shapley stable marriage method gives the stable values but not the preferred ones. The improved Hungarian method guarantees the QoS provided to GAA users and also ensures that the Idle PAL reserved channels are successfully allocated to GAA users while satisfying the FCC rules. The interference experienced by GAA users remains under the threshold limit if only one channel is allocated to GAA users.

## 6. Conclusions

In order to meet the requirements of licensed and un-licensed users, FCC proposed to commercialize the CBRS band held by the government. There is a need to define methods to assign the 3.5 GHz CBRS band. In this regard, FCC proposed SAS based architecture to provide services to three types of users including IA users, PAL users and GAA users. FCC also proposed a set of rules to assign channels and provide guaranteed QoS to IA users and PAL users; however, no interference protection is guaranteed to GAA users. Hence, a method that complies with the FCC rules to allocate channels to GAA users is required to guarantee interference protection to the PAL users yet improve the QoS for the GAA users. In this paper, an algorithm to assign channels to GAA users satisfying the FCC rules is proposed. The channel allocation method proposed in this method is modelled as a multi-objective optimization problem with the goals of maximizing the transmission rate of GAA users at minimum cost subject to interference and QoS constraints based on FCC rules. The NP hard channel allocation model is transformed to a 0–1 integer programming problem using a simplification method. The problem is then solved by using the proposed improved Hungarian method and also solved by the classical Hungarian method, Gale–Shapley stable matching problem and the auction algorithm integrating interference and QoS constraints.

Simulations are performed that show that the Gale–Shapley stable matching algorithm yields the stable pairs but not the optimal data rate per unit cost values. The classical Hungarian algorithm, auction algorithm and improved Hungarian algorithm converge to the same values and result in optimal data rate per unit cost. Therefore, computational complexity of an algorithm is the deciding factor for SAS to use it for assigning the Idle PAL reserved channels to GAA users. Results show that, at a higher network load, the improved Hungarian method is assigning channels and achieving maximum transmission rate at a minimum cost satisfying FCC rules with approximately up to four times better execution efficiency. The interference experienced by GAA users remains under threshold limits if, on average, one channel is assigned to each GAA user.

The research work includes important methods and results that can be used for further studies. Moreover, this research work can be extended to assign multiple channels to GAA users using co-channel interference models to improve the interference experienced by GAA users, in order to integrate multiple spectrum access systems and sharing resources among them incorporating IA users, PAL users and GAA users.

## Figures and Tables

**Figure 1 sensors-22-01318-f001:**
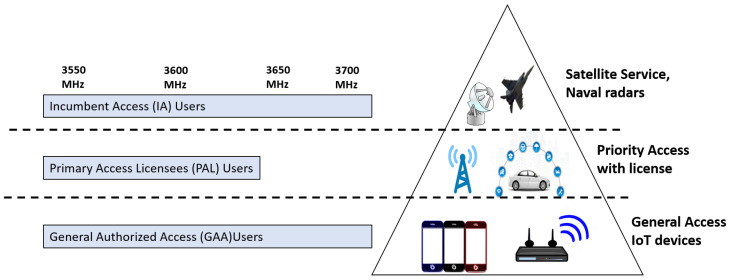
Layered architecture.

**Figure 2 sensors-22-01318-f002:**
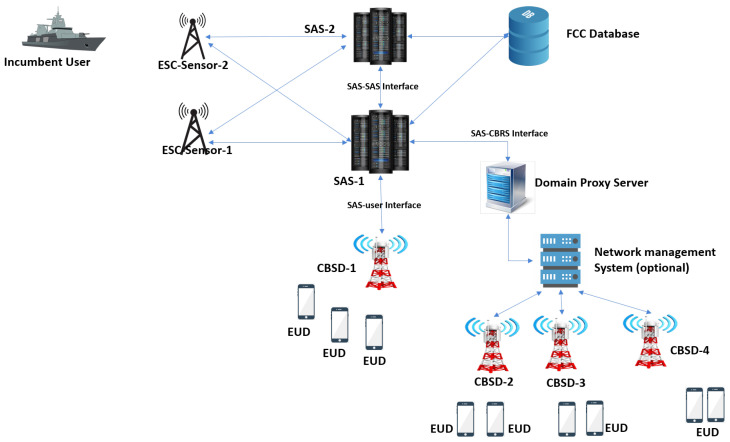
CBRS-SAS architecture.

**Figure 3 sensors-22-01318-f003:**
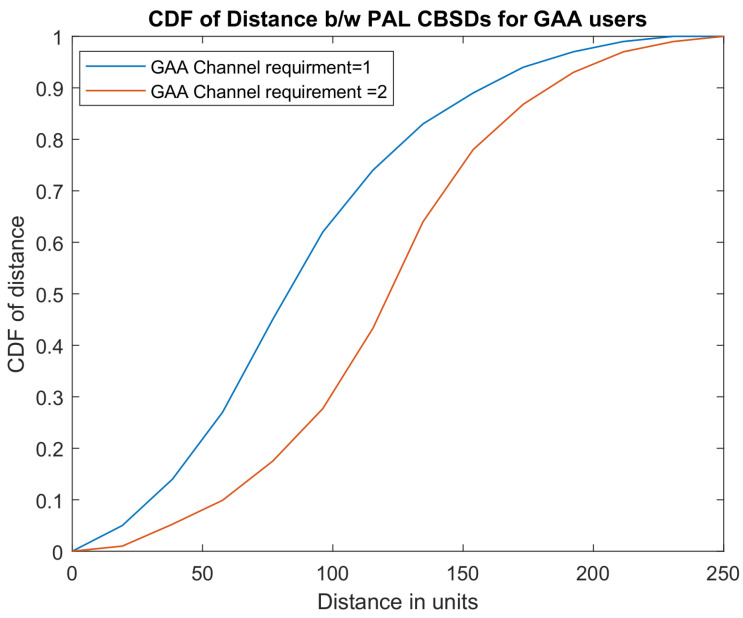
CDF of InterCBSD distance.

**Figure 4 sensors-22-01318-f004:**
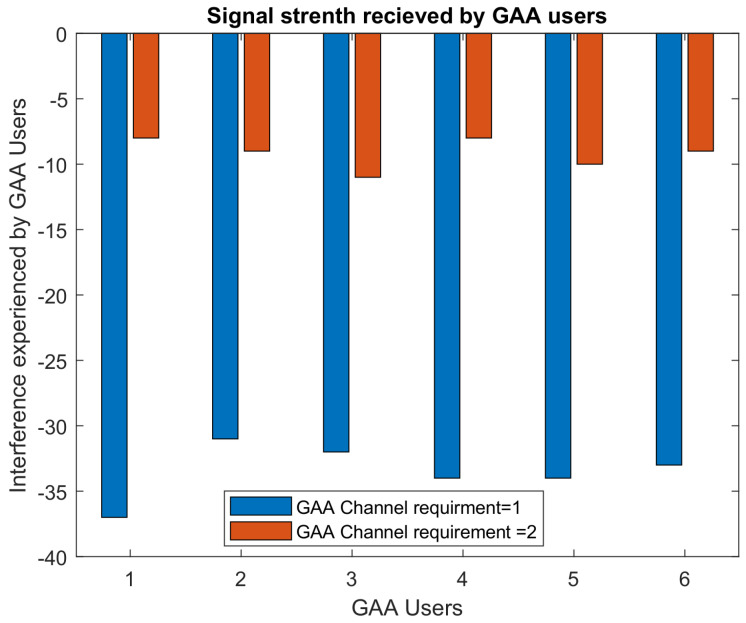
Signal strength received by GAA users.

**Figure 5 sensors-22-01318-f005:**
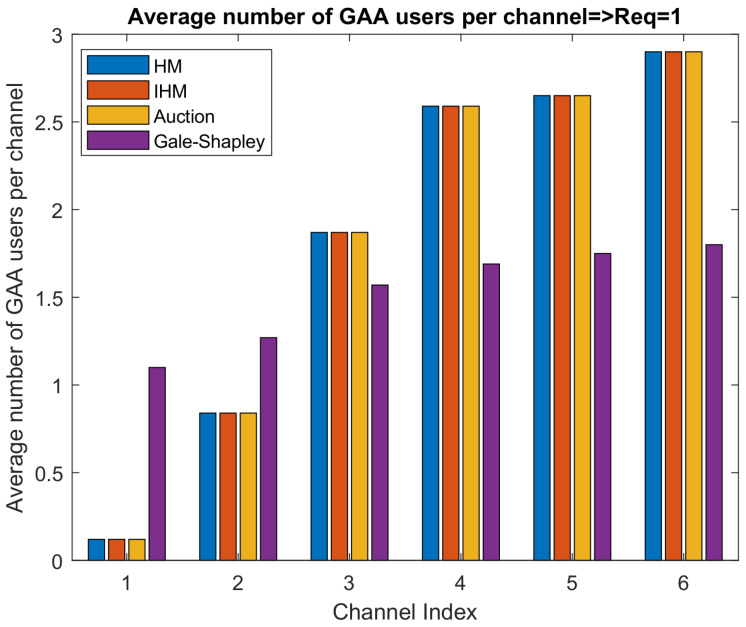
Average number of users when channel required: 1.

**Figure 6 sensors-22-01318-f006:**
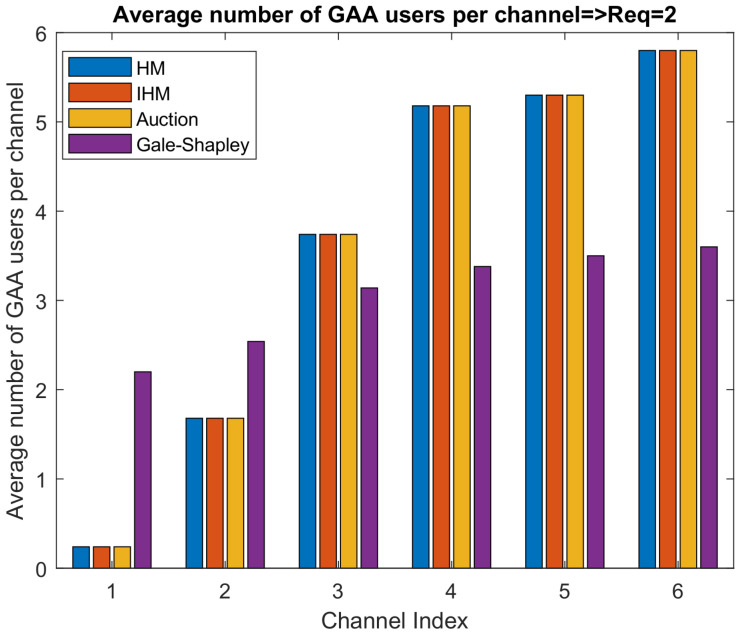
Average number of users when channels required: 2.

**Figure 7 sensors-22-01318-f007:**
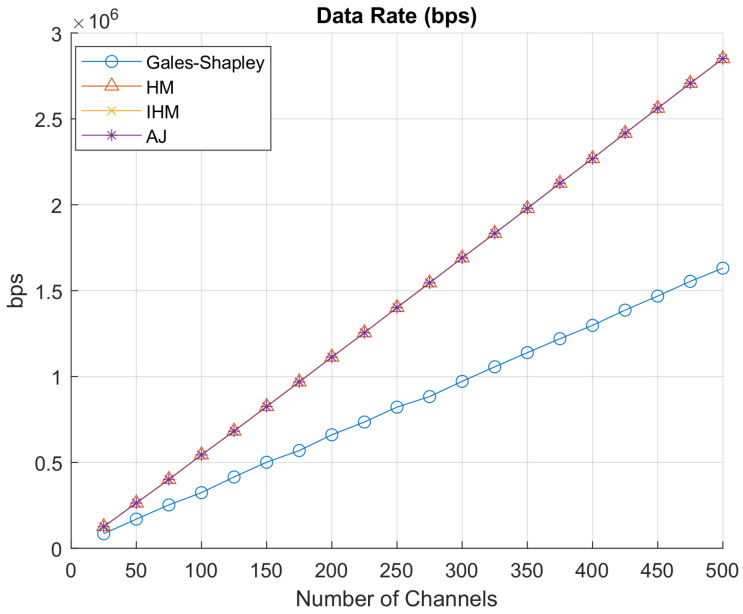
Net datarate.

**Figure 8 sensors-22-01318-f008:**
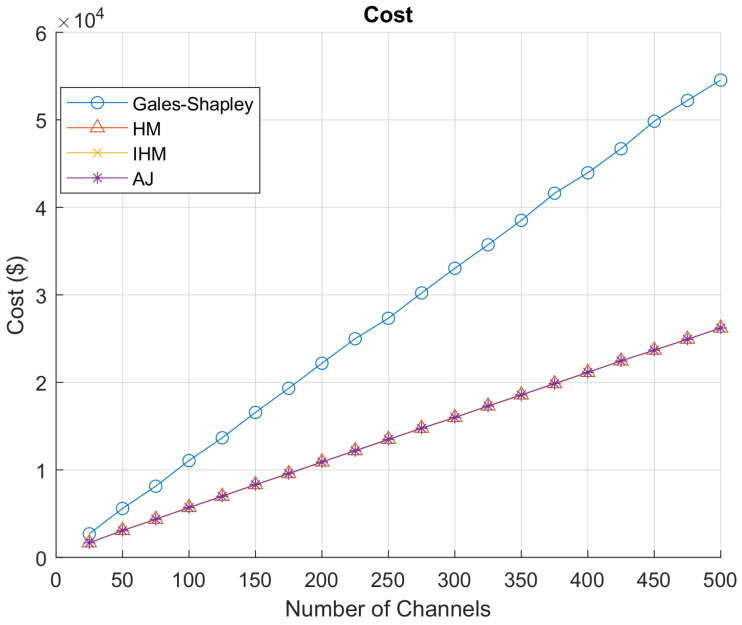
Net cost.

**Figure 9 sensors-22-01318-f009:**
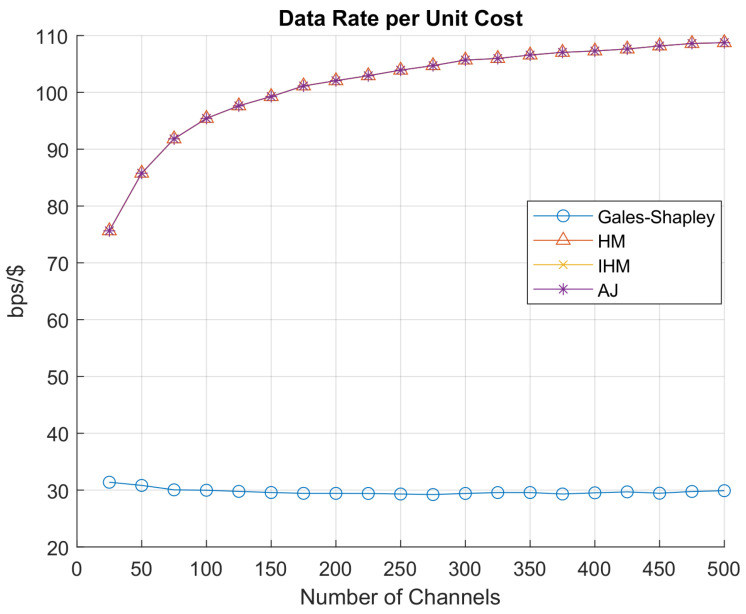
Datarate per unit cost.

**Figure 10 sensors-22-01318-f010:**
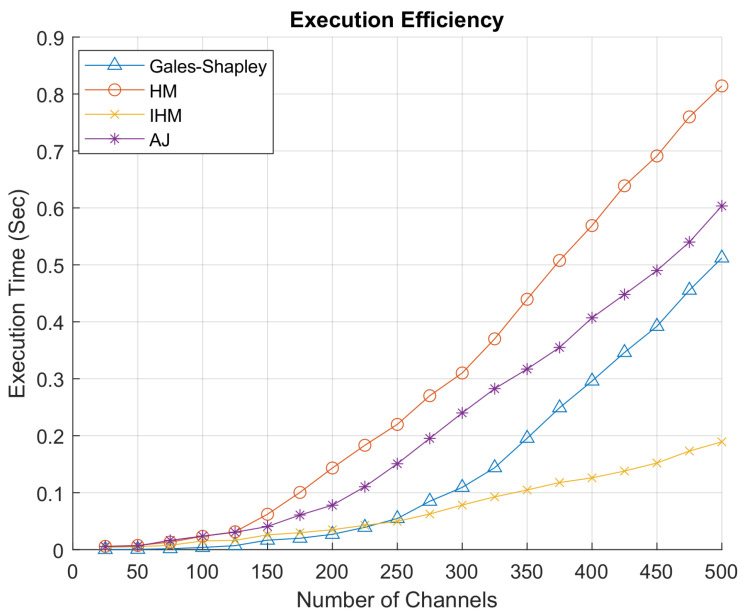
Execution efficiency.

**Table 1 sensors-22-01318-t001:** Notations Used.

Set	Description
*I*	Set of incumbent users
*P*	Set of PAL users
*G*	Set of GAA users
$S_{p}$	Set of PAL reserved channels
$S_{g}$	Set of GAA reserved channels
$S_{m}$	Idle PAL reserved channel allocated to GAA user *g*
*S*	Set of all channels
*S*	Set of all channels

**Table 2 sensors-22-01318-t002:** Idle spectrum assignment.

GAA	Idle PAL Reserved Channels $S_{p}$								
Users	$S_{1}$	$S_{2}$	$S_{3}$	$S_{4}$	$S_{5}$	$S_{6}$	...	$S_{m-1}$	$S_{m}$
$G_{1}$	0	1	0	0	0	0	...	0	0
$G_{2}$	1	0	0	0	0	0	...	0	0
$G_{3}$	0	0	1	0	0	0	...	0	0
$G_{4}$	0	0	0	1	0	0	...	0	0
$G_{5}$	0	0	0	0	0	1	...	0	0
$G_{6}$	0	0	0	0	1	0	...	0	0
...	.........								
$G_{g-1}$	0	0	0	0	0	0	...	1	0
$G_{g}$	0	0	0	0	0	0	...	0	1

**Table 3 sensors-22-01318-t003:** Bandwidth efficiency matrix.

GAA	Bandwidth Efficiency Matrix								
Users	$S_{1}$	$S_{2}$	$S_{3}$	$S_{4}$	$S_{5}$	$S_{6}$	...	$S_{m-1}$	$S_{m}$
$G_{1}$	$b_{11}$	$b_{12}$	$b_{13}$	$b_{14}$	$b_{15}$	$b_{16}$	...	$b_{1(m-1)}$	$b_{1m}$
$G_{2}$	$b_{21}$	$b_{22}$	$b_{23}$	$b_{24}$	$b_{25}$	$b_{26}$	...	$b_{2(m-1)}$	$b_{2m}$
$G_{3}$	$b_{31}$	$b_{32}$	$b_{33}$	$b_{34}$	$b_{35}$	$b_{36}$	...	$b_{3(m-1)}$	$b_{3m}$
$G_{4}$	$b_{41}$	$b_{42}$	$b_{43}$	$b_{44}$	$b_{45}$	$b_{46}$	...	$b_{4(m-1)}$	$b_{4m}$
$G_{5}$	$b_{51}$	$b_{52}$	$b_{53}$	$b_{54}$	$b_{55}$	$b_{56}$	...	$b_{5(m-1)}$	$b_{5m}$
$G_{6}$	$b_{61}$	$b_{62}$	$b_{63}$	$b_{64}$	$b_{65}$	$b_{66}$	...	$b_{6(m-1)}$	$b_{6m}$
...	.........								
$G_{g-1}$	$b_{(g-1)1}$	$b_{(g-1)2}$	$b_{(g-1)3}$	$b_{(g-1)4}$	$b_{(g-1)5}$	$b_{(g-1)6}$	...	$b_{\left(g-1)(m-1\right)}$	$b_{(g-1)m}$
$G_{g}$	$b_{g}1$	$b_{g2}$	$b_{g3}$	$b_{g4}$	$b_{g5}$	$b_{g6}$	...	$b_{\left(g)(m-1\right)}$	$b_{gm}$

**Table 4 sensors-22-01318-t004:** Cost for GAA users accessing the idle spectrum.

GAA	Cost for GAA Users Accessing the Idle Spectrum								
Users	$S_{1}$	$S_{2}$	$S_{3}$	$S_{4}$	$S_{5}$	$S_{6}$	...	$S_{m-1}$	$S_{m}$
$G_{1}$	$c_{11}$	$c_{12}$	$c_{13}$	$c_{14}$	$c_{15}$	$c_{16}$	...	$c_{1(m-1)}$	$c_{1m}$
$c_{2}$	$c_{21}$	$c_{22}$	$c_{23}$	$c_{24}$	$c_{25}$	$c_{26}$	...	$c_{2(m-1)}$	$c_{2m}$
$G_{3}$	$c_{31}$	$c_{32}$	$c_{33}$	$c_{34}$	$c_{35}$	$c_{36}$	...	$c_{3(m-1)}$	$c_{3m}$
$G_{4}$	$c_{41}$	$c_{42}$	$c_{43}$	$c_{44}$	$c_{45}$	$c_{46}$	...	$c_{4(m-1)}$	$c_{4m}$
$G_{5}$	$c_{51}$	$c_{52}$	$c_{53}$	$c_{54}$	$c_{55}$	$c_{56}$	...	$c_{5(m-1)}$	$c_{5m}$
$G_{6}$	$c_{61}$	$c_{62}$	$c_{63}$	$c_{64}$	$c_{65}$	$c_{66}$	...	$c_{6(m-1)}$	$c_{6m}$
...	.........								
$G_{g-1}$	$c_{(g-1)1}$	$c_{(g-1)2}$	$c_{(g-1)3}$	$c_{(g-1)4}$	$c_{(g-1)5}$	$c_{(g-1)6}$	...	$c_{\left(g-1)(m-1\right)}$	$c_{(g-1)m}$
$G_{g}$	$c_{g1}$	$c_{g2}$	$dc_{g3}$	$c_{g4}$	$c_{g5}$	$c_{g6}$	...	$c_{\left(g)(m-1\right)}$	$c_{gm}$

**Table 5 sensors-22-01318-t005:** Relative bandwidth-cost efficiency matrix.

GAA	Relative Banwidth-Cost Efficiency Factor								
Users	$S_{1}$	$S_{2}$	$S_{3}$	$S_{4}$	$S_{5}$	$S_{6}$	...	$S_{m-1}$	$S_{m}$
$G_{1}$	$\alpha_{11}$	$\alpha_{12}$	$\alpha_{13}$	$\alpha_{14}$	$\alpha_{15}$	$\alpha_{16}$	...	$\alpha_{1(m-1)}$	$\alpha_{1m}$
$G_{2}$	$\alpha_{21}$	$\alpha_{22}$	$\alpha_{23}$	$\alpha_{24}$	$\alpha_{25}$	$\alpha_{26}$	...	$\alpha_{2(m-1)}$	$\alpha_{2m}$
$G_{3}$	$\alpha_{31}$	$\alpha_{32}$	$\alpha_{33}$	$\alpha_{34}$	$\alpha_{35}$	$\alpha_{36}$	...	$\alpha_{3(m-1)}$	$\alpha_{3m}$
$G_{4}$	$\alpha_{41}$	$\alpha_{42}$	$\alpha_{43}$	$\alpha_{44}$	$\alpha_{45}$	$\alpha_{46}$	...	$\alpha_{4(m-1)}$	$\alpha_{4m}$
$G_{5}$	$\alpha_{51}$	$\alpha_{52}$	$\alpha_{53}$	$\alpha_{54}$	$\alpha_{55}$	$\alpha_{56}$	...	$\alpha_{5(m-1)}$	$\alpha_{5m}$
$G_{6}$	$\alpha_{61}$	$\alpha_{62}$	$\alpha_{63}$	$\alpha_{64}$	$\alpha_{65}$	$\alpha_{66}$	...	$\alpha_{6(m-1)}$	$\alpha_{6m}$
...	.........								
$G_{g-1}$	$\alpha_{(g-1)1}$	$\alpha_{(g-1)2}$	$\alpha_{(g-1)3}$	$\alpha_{(g-1)4}$	$\alpha_{(g-1)5}$	$\alpha_{(g-1)6}$	...	$\alpha_{\left(g-1)(m-1\right)}$	$\alpha_{(g-1)m}$
$G_{g}$	$\alpha_{g}1$	$\alpha_{g2}$	$\alpha_{g3}$	$\alpha_{g4}$	$\alpha_{g5}$	$\alpha_{g6}$	...	$\alpha_{\left(g)(m-1\right)}$	$\alpha_{gm}$

**Table 6 sensors-22-01318-t006:** Relative bandwidth–cost efficiency matrix.

GAA	New Relative Banwidth-Cost Efficiency Factor								
Users	$S_{1}$	$S_{2}$	$S_{3}$	$S_{4}$	$S_{5}$	$S_{6}$	...	$S_{m-1}$	$S_{m}$
$G_{1}$	$\gamma_{11}$	$\gamma_{12}$	$\gamma_{13}$	$\gamma_{14}$	$\gamma_{15}$	$\gamma_{16}$	...	$\gamma_{1(m-1)}$	$\gamma_{1m}$
$G_{2}$	$\gamma_{21}$	$\gamma_{22}$	$\gamma_{23}$	$\gamma_{24}$	$\gamma_{25}$	$\gamma_{26}$	...	$\gamma_{2(m-1)}$	$\gamma_{2m}$
$G_{3}$	$\gamma_{31}$	$\gamma_{32}$	$\gamma_{33}$	$\gamma_{34}$	$\gamma_{35}$	$\gamma_{36}$	...	$\gamma_{3(m-1)}$	$\gamma_{3m}$
$G_{4}$	$\gamma_{41}$	$\gamma_{42}$	$\gamma_{43}$	$\gamma_{44}$	$\gamma_{45}$	$\gamma_{46}$	...	$\gamma_{4(m-1)}$	$\gamma_{4m}$
$G_{5}$	$\gamma_{51}$	$\gamma_{52}$	$\gamma_{53}$	$\gamma_{54}$	$\gamma_{55}$	$\gamma_{56}$	...	$\gamma_{5(m-1)}$	$\gamma_{5m}$
$G_{6}$	$\gamma_{61}$	$\gamma_{62}$	$\gamma_{63}$	$\gamma_{64}$	$\gamma_{65}$	$\gamma_{66}$	...	$\gamma_{6(m-1)}$	$\gamma_{6m}$
...	.........								
$G_{g-1}$	$\gamma_{(g-1)1}$	$\gamma_{(g-1)2}$	$\gamma_{(g-1)3}$	$\gamma_{(g-1)4}$	$\gamma_{(g-1)5}$	$\gamma_{(g-1)6}$	...	$\gamma_{\left(g-1)(m-1\right)}$	$\gamma_{(g-1)m}$
$G_{g}$	$\gamma_{g}1$	$\gamma_{g2}$	$\gamma_{g3}$	$\gamma_{g4}$	$\gamma_{g5}$	$\gamma_{g6}$	...	$\gamma_{\left(g)(m-1\right)}$	$\gamma_{gm}$

**Table 7 sensors-22-01318-t007:** Characteristics of PAL network operators.

PAL Operators	Available Channels	Bandwidth (Kbps)	Cost	Packet Loss	Delay (ms)	Interference
PAL1	10	5500–6000	130–150	<1	50–60	−30 dB
PAL2	10	1500–2000	80–100	<1	40–45	−30 dB
PAL3	10	1000–1200	60–80	<1	40–45	−30 dB
PAL4	10	3000–3500	140–160	<1	50–60	−30 dB

**Table 8 sensors-22-01318-t008:** GAA users’ classification and their QoS requirements.

GAA Users	Transmission Rate	Cost	Packet Loss	Delay (ms)	Interference
GAA-1	2000–2500	100	<2	55	−40
GAA-2	500–1000	60	<3	50	−40
GAA-3	1500–2000	100	<1	45	−40

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
