# Peer review of "Resource Allocation in Spectrum Access System Using Multi-Objective Optimization Methods"

_sensors, 2022, doi:10.3390/s22041318_

Round 1

Reviewer 1 Report

The paper proposes a dynamic shared access model for the citizen broadband radio service (CBRS) band, based on multi-objective optimization. The proposed mathematical model optimizes the selection of PAL operators and idle PAL reserved channels allocation to GAA users,  considering the diversity of PAL reserved channels attributes and the diversification of GAA users business need. Various performance metrics and  several exsiting algorithms have been involved in the experiments. The experimental results indicates that the key performance of the proposed method remains the same as the state-of-the-art methods but four times faster.

Overall, the paper is well-written and easy to follow. The insights are well-presented. The topic of resource allocation in spectrum access system is hot and interesting. The comparison with Hungarian algorithm, auction algorithm and Gale-Shapley algorithm is quite impressive. However, it would be better, if there is more description about the Figure 1 or more background information is introduced. Another minor issue is that all the figures are somehow blur. Please use figures with higher resolution.

Author Response

response letter attached.

Reviewer 2 Report

This paper presents a new mathematical model based on multi-objective optimization for allocating channels with maximum transmission rate and minimum cost, where different algorithms are used. The results of the proposed model are analyzed, compared, and validated with previous work achieved. The authors have been done great work, and thanks to this effort. However, minor revisions are required in the paper to warrant further consideration for publication.

  • The main contributions made are not outlined, and why the need for this work is not clear.
  • The authors should carefully double-check the definition of abbreviations for those terms throughout the manuscript, including WIFI in line 48, WIN in line 101, CRN in line 140, EIRP in line 154, IoT in line 158, and others.
  • I suggest for authors add more recent papers (e.g. 2020, 2021, and 2022 if available) in the related works section.
  • In references, authors should add the name of city and country for all cited conferences papers.

Author Response

response letter attached.

Reviewer 3 Report

Rephrase "less iterations and takes much less time". Express the numeric values of the improvements attained.
01 programming and 0-1 programming - Please standardize the phrase.
"strenth" - Typo
"Linear assignment problem (LAP) simplification method proposed in this article is the improved version of classical Hungarian algorithm." - Please name the proposed method properly - is it LAP method, improved Hungarian algorithm, or modified Hungarian algorithm?
Explain the differences in performance shown in Figs. 4 and 5.
Explain the very different performance of the Gale-Shapley method as compared to others in Figs. 6-8.
The results and conclusion are not convincing.

Author Response

response letter attached.
